# The Impacts of COVID-19 on Technological and Polytechnic University Teachers

**Lourdes Vital-López** [1], **Raul García-García** [2], **Juvenal Rodríguez-Reséndíz** [3],
**Willfrido Jacobo Paredes-García** [3], **Marco Antonio Zamora-Antuñano** [4], **Temidayo Oluyomi-Elufisan** [1],
**Hugo Rodríguez Reséndiz** [3], **Ana Ruth Álvarez Sánchez** [5] and **Miguel Angel Cruz-Pérez** [4,*]

1  Carrera de Mantenimiento Industrial, Universidad Tecnológica de Tamaulipas Norte,
   Universidad Tecnológica, Reynosa 88680, Mexico; lourdes.vital@uttn.mx (L.V.-L.);
   ptemidayo@gmail.com (T.O.-E.)
2  División de Química y Energías Renovables, Universidad Tecnológica de San Juan del Río,
   Querétaro 76800, Mexico; rgarciag@utsjr.edu.mx
3  Facultad de Ingeniería, Universidad Autónoma de Querétaro, Querétaro 76010, Mexico;
   juvenal@uaq.edu.mx (J.R.-R.); wparedes17@alumnos.uaq.mx (W.J.P.-G.); hugorore@uaq.mx (H.R.R.)
4  Posgrado y Centro de Investigación e Innovación Tecnológica (CIIDETEC-UVM), Universidad del Valle de
   Mexico, Querétaro 76230, Mexico; marco.zamora@uvmnet.edu
5  Postgraduate Unit, Universidad Técnica Estatal de Quevedo, Los Rios, Quito 76230, Ecuador;
   aalvarezs@uteq.edu.ec
*  Correspondence: miguel_cruzp@my.uvm.edu.mx

**Abstract:** University teachers have adapted to different situations during the development of distance learning due to the pandemic caused by the COVID-19 virus. This study was conducted by assigning a data collection instrument to 993 teachers who are part of 15 technological universities (TUs) and 7 polytechnic universities (PUs) to determine how they were affected by COVID-19. The questions asked were related to the social, economic, academic, emotional, and health effects experienced. The results show that 63% of the teachers working online complained that online teaching invaded their family privacy; 56% pointed out that working from home and the virtual classes affected their performance as teachers; 90% of the teachers thought that they dedicated too much extra to preparing for their classes; 15% were stressed; 4% felt negative under the new teaching scheme of virtual classes; finally, 38% of the teachers stated that repeated interaction with electronic devices had a lot of negative impacts on their emotional wellbeing. By means of a G-test, it was determined that gender was independent from the studied effects. Through a multiple correspondence analysis (MCA), it was determined that, of the total number of teachers who responded to the questionnaire, half were comfortable with the online teaching model and the other half were not. The most impacted effects were the economic, training and connectivity independently from the gender.

**Keywords:** COVID-19; pandemic; teachers; universities; faculty members; impact; shifts

## 1. Introduction

Since December 2019, COVID-19 has spread rapidly from China to many countries around the world because of international travel, and it became one of the global challenges placing a significant burden on the healthcare sector. The COVID-19 pandemic has affected many countries globally, resulting in the implementation of strict control measures to stop the viral spread. The closure of universities and the adoption of online learning were among these preventive measures [1,2]. UNESCO reported university closures in more than 160 countries that are not sure of how long the coronavirus crisis will last and how it could affect the mental health of the students and faculty [3]. The psychological effects of the sudden transition from face-to-face to online classes generated anguish and uncertainty [4]. Home confinement as a measure to avoid contagion also made many teachers feel afraid, hopeless, and stressed [5]. The closure of universities was a protective measure considered

by studies comparing the impact of the spread of COVID-19 among those exposed in face-to-face teaching and unexposed teachers (online teaching), which revealed that the spread of the virus in the exposed community doubled compared to the unexposed teachers. In addition, it was also shown that those in the exposed categories also serve as a vehicle of transmission to others—mostly their partners. These studies concluded that adequate protection should be provided for the teachers; however, this has led to a high degree of social isolation among university staff and students [6,7]. The online teaching method was adapted to keep education going while not exposing the teachers to infection [8]. However, the adoption of the online or distance learning scheme opened a new chapter in the understanding of the mental health stability of teachers and students. Online education is not limited to distance education, as it refers to a grouping of teaching–learning procedures completed in cyberspace [9–11]. Mental health problems can have a negative impact on the physical and psychological wellbeing of students and predispose them to many unhealthy behaviors [8]. Melaku (2021) found that depression, anxiety, and stress were common problems among Arsi University medical students exposed to online teaching activities [12]. Gregori (2021) evaluated a program in which teachers served as coaches for professionals [13]. The results showed that with teacher coaching, paraprofessionals increased their behavior intervention implementation fidelity to a hundred percent. The teaching roles in the time of the pandemic include offering support for home confinement, promoting resilience, academic guidance, preventing procrastination, empathetic and active listening, emotional and institutional advising, and acting as a motivator [5,14]. Other requirements for teachers include voice modulation, becoming familiar with the equipment and software and optimizing its use, being mindful of body language, using a script to optimize the use of time, and allowing the students to express themselves freely. During the execution of the classes, teachers are recommended to give clear instructions, resolve doubts, and explain the projects and tasks, among others. In order to fulfill these roles and requirements, the mechanism for evaluating the students and assignment deadlines should be clearly stated and, as a suggestion, tutorials should have clear explanations [5,8].

Fedock (2019) explored how adjunct higher education faculty perceive using social media (SM) as an instructional tool for their students during the pandemic and found out that it is one of the most effective tools for knowledge dissemination [15]. In other related studies, the reliability and effectiveness of different information technology (IT) tools used by teachers were evaluated and it was found that most of the knowledge transfer was based on learning tools (LT), the use of mobile instruments (such as cell phones and tablets), and virtual libraries (VL), with levels of 89%, 85%, and 82%, respectively [16]. In addition, they found that the use of cell phones made the teaching–learning process more dynamic. Adopting an online learning approach has proven to be an alternative to physical classrooms in an uncontrollable situation. This has allowed universities, faculty, and students to have patience and resilience, which will be useful for future challenges in high-quality education [17–19]. Another effect of the pandemic is a diminishment in the quality of education that all students deserve [20]; the absence of online learning infrastructure could have worsened the situation worldwide. The physical distancing amid the pandemic has influenced the attitudes of teachers worldwide opting for social media (SM) use in online learning, mainly in developing countries. Switching to online learning using SM under challenging situations like the COVID-19 pandemic is, thus, inevitable [21,22]. Many students have lost close family members and must continue to study under these conditions. In Mexico, higher education institutions (HEIs) are classified into six large groups: public universities, technological education, technological and polytechnic universities, private institutions, normal education, and other public institutions. All the categories switched to the virtual mode of teaching during the pandemic.

In this study, we explored the effects of COVID-19 on the social, economic, academic, emotional, and health aspects of the lives of teachers at the TUs and PUs of Mexico during the period of March to August 2021. The General Coordination of Technological and Polytechnic Universities (GCTPU) is a part of the Ministry of Education in Mexico

(SEP). The GCTPU is part of the secretariat or higher education that changed its name in 2018, originally being the General Directory of Technological and Polytechnic Universities (UUTT). The coordination works as a scheme of higher education to fulfill the requirements of the society and students integrated into the productive sector with a committed and consolidated teaching team. This system seeks national and international recognition for its efficiency, effectiveness, relevance, equity, and linkage. It is open, flexible, innovative, integrated into the other subsystems of higher education, and linked to the social and productive sectors that contribute to the economic development of the country, in culture, science, and technology. The academic offerings are equivalent to the community colleges of the higher technical universities in the United States and Canada, and to the university institutes of technology (IUT) in France [23]. These are university systems that prepare technicians to be able to immediately enter the labor market or to continue with higher studies. The educational model has a specific formation scheme 30% is theory-based and 70% is practice-based education. This model was established in Mexico in 2001 to address professional and qualification needs. It allows for future technicians, graduates, and engineers with a higher level of education to enter the productive sector with programs known as professional internships. In 2021, there were 114 technological universities and 62 polytechnic universities in Mexico. The GCTPU serves 300,000 students, which represents 7.5% of the total amount of students (4 million) in higher education in Mexico [24]. The objective of this study to specifically study the GCTPU teachers has not been applied to other higher education institutions. The GCTPU teachers follow a specific model given the 70% practical and 30% theoretical distribution of the syllabus. Other studies [25] have had different perspectives on the virtualization of the teaching–learning process during the pandemic. Gender and age seem to be important factors in teacher satisfaction, so it is of interest to find out if that behavior repeats in the specific model of the GCTPU while using the virtual learning platforms (VLPs) recommended by the institutions. This study is important for reinforcing the need for public HEIs because most of the studies of this kind have been about private HEIs, with a focus on other teachers behaving differently. Modgil (2019) explored how adjunct higher education faculty perceive using SM as instructional tool for their students during the pandemic, and found out that it is one of the most effective tools for knowledge dissemination [26]. In other related studies they evaluated the reliability and effectiveness of different information technology (IT) tools for knowledge transfer by teachers and found that most of the knowledge transfer by teachers was based on learning tools (LT), the use of mobile instruments (such as cell phones and tablets) and virtual library (VL), with 89%, 85% and 82%, respectively [16]. In addition, they found that the use of cell phones made the teaching–learning process more dynamic [15]. Adopting online–learning approach has proven to be the alternative to physical classroom in an uncontrollable situation. This has allowed universities, faculty, and students to have patience and resilience, as they will be useful for future challenges in high–quality education [17–19].

Another effect of the pandemic is a quality diminishment in the education all students deserve the absence of online learning infrastructure could have worsened the situation worldwide [20]. The physical distancing amid the pandemic has influenced the attitudes of the teachers worldwide opting for social media (SM) use in online learning, mainly in developing countries; switching to online learning using SM under challenging situations like the COVID-19 pandemic is thus inevitable [21,22]. Many students have lost close families, and must continue to study under these conditions. In Mexico, HEIs are classified in six large groups: public universities, technological education, technological and polytechnic universities, private institutions, normal education, and other public institutions [24,27]. All the categories switched to virtual mode of teaching during the pandemic. In this study we explore COVID-19 effects on the social, economic, academic, emotion and health of teachers of the TUs and PUs of Mexico, during the period of March to August 2021 [24].

### 1.1. Theoretical Framework

There are several challenges that teachers have had in the face of the COVID-19 pandemic; most have declared that their actions were focused not only on the academic component but also on the emotional [28]. Teachers, being on the front lines with their students, were probably not trained to respond to the threats to the emotional wellbeing of themselves and their students, and they may have experienced situations of stress and anxiety [29]. Different investigations on the psychological reactions resulting from the pandemic have suggested that some factors related to anxiety or health had an impact on the vulnerability of some teachers. Among some situations that can be mentioned are a tolerance to uncertainty, the self-perception of their susceptibility to the disease, and anxiety [30]. Other factors, such as being female, having COVID-19 symptoms, misinformation, social isolation, low educational levels, unemployment, or losing one's job were the main situations that had the reported greatest psychological effects and seem to have been associated with the highest levels of anxiety and depression [31]. Research in China found that young women were among the most vulnerable groups to mental health consequences from the pandemic, as many working women also took care of their homes [32]. Therefore, the closure of school institutions can substantially reduce job performance for fear of not maintaining employment, limiting development opportunities, and affecting one's financial status [28]. However, and paradoxically, the pandemic has also motivated teachers to have greater commitment to their academic work, and some teachers even perceive that the situation of confinement allowed them to spend more time with their families and have greater comfort and reduced their travel expenses [33,34]. In the case of teachers, the new dynamics of teaching and learning made their work more intense, and it overflowed into their personal time and lives [35]. The need to undergo continuous training predisposed them to the abandonment of leisure activities, sport, and other practices that reduce stress [36]. Alavudeen and Easwaran (2021) generated affectations such as a lack of experience or ignorance in the management of virtual learning platforms (VLPs), poor connectivity, and poor audio/video quality were impotent barriers that increased psychological stress, study discipline, and life status that caused a non-positive impact on the performance of the teaching–learning process in online education [37]. Psychological distress, technical problems associated with accessibility, inexperience, and a lack of preparation were found to be the main barriers limiting students' acceptance of learning that impacted the outcome of their assessments Mexico was not the exception, as COVID-19 altered the work of HEIs in Mexico, resulting in both expected and unexpected consequences that are yet to be identified and evaluated. Many HEIs undertook different actions and studies to determine the short- and medium-term implications [9,12,17,38]. A preliminary analysis indicated that, in general, HEIs did not have any foresight to face the crisis and its arrival took the HEIs by surprise. However, they reacted positively both to contribute to the reduction of the risks of contagion and to resume their functions and fulfill their commitments. Important institutional processes have been affected even without knowing the negative impacts of the short and medium term. The technological universities have undertaken analysis projects on the impact of COVID-19 and the effects on their academic community of teachers and students. The present work presents the results of one of the first analyses of the situation of COVID-19, confinement, and the virtualization of processes at this educational level. Not all the different entities that make up the Ministry of Education in Mexico (SEP) undertook actions of this nature, which makes the results achieved a project of value for the establishment of systems to improve the teaching–learning process in a virtual way [24]. This research is innovative because an analysis of the health of teachers in technological and polytechnic universities was not carried out during distance education activities due to COVID-19, and it allows for studying the problems, contributes updated data, and projects future solutions. It also has social relevance because it highlights the causes and effects of the problems on staff and educational institutions, supporting decision making and the improvement of the conditions of distance education. The main aspects are described within this research paper. Below are the objectives.

Gender Impact during COVID-19

Gender is an important issue as a trending topic and more during the pandemic. There are important advances in this regard throughout recent years such as technology improvements, but there are some challenges to face, such as gender equity in access, in digital devices ownership, in training for digital fluency, and ability accessing technology. Although affordability is key factor for exclusion, the analysis of that matter should be studied specific working environments and working opportunities, but also in financial inclusion. Technology represents an omnipresent element that affects globally and internet assumes incorporating the individual to a interconnected society where inclusion represents a competitive advantage in development, integration and wellness [39,40].

Imbalanced coverage in connectivity, technology appropriation generates a digital gap between those with access and those without coverage. This gap could be attributed to a geographic, economic, cultural, and generational disparity. Alva (2015) declared the presence of a digital gap in three dimensions: access, use and appropriation. He explained that these dimensions give three particular gaps: (a) digital gap of use, (b) age range digital gap between native and digital inmigrants, and (c) gender digital gap [41].

The digital gap could be attributed to diverse factors. According to Instituto Nacional de Estadística, Geografía e Informática de Mexico (INEGI), the factors are school attendance, being ages 15–17 predominantly male; education level, where female 15+ is lower than the years attending to school by 15+ males; lower participation of female (23.7%) in Science, Technology, Engineering and Mathematics (STEM); cultural matters as reading habits, which is lower in female 25+; low attendance of female to cultural activities (39.8%) promoting their personal development; and a lower economic participation by sector in the country which is of 95 per 100 male in population in the age range 30–49. These factors maintain a rough affectation, mainly to female, which stood up during the COVID-19 pandemic: 66.6% of female 12+ work 30.8 h to non paid activities their need to work and being in charge of the family and house keeping or by being pregnant, which make their activity a non paid one, while male use 28% of their time, or 11.6 h to those activities [42].

Korlat (2021) studied four components of digital learning that are susceptible of stereotyped gender gap [43]. While Lawal et al. (2021) did not find significance differences regarding gender, but it is attributed to the fact that both genders were submitted to similar COVID-19 protocols during the pandemic [44]. Female experience unique health risks resulting from their gender. Many of these studies identified inequity in the academic world for female [45]. The barriers include disparity in economic compensation and inequity in the three pillars of academic assessment: teaching, service and research. Due to uneven payment for female teachers, specially those non-white, it is highly probable that the stressing economic factors were exacerbated during the pandemic, particularly for homes with a female as head of the family or single mothers, and predominantly for those in incidental teaching positions [45]. Regarding scientific literature produced before the pandemic, Garduño (2020) considers that teacher exposition to psychosocial risks derived from their exposition to the school environment is also a future possibility for distance teaching. This implies a greater exposition to depression, stress and mental health issues. Literature also points out the relationship between inadequate working conditions and psychosocial consecuences, sucha as stress, dysphonia and voice related problems, phisical inactivity during free time and anxiety. This reality could also be be different for male and female in the labor market. From this differences, literature centers its attention in the higher exposition of female to domestic violence due to lockdowns and in the working environment it is legitimate to consider that female could be overwhelmed [46].

## 1.2. Objectives

The objectives of this research are as follows: To identify the sociodemographic aspects of the PU and TU teachers that have an impact on the use of virtual learning platforms (VLPs); to learn the perceptions of teaching staff at the higher education institutions (HEIs) both technological and polytechnical universities (TUs and PUs)—regarding the change

from face-to-face to VLP during the COVID-19 crisis; to identify the main problems that teachers faced in the process of virtualization of teaching–learning and how this contributes to the strategies in preparing the educational institutions for this change.

*1.3. Research Questions and Research Hypothesis*

This study seeks to identify the impacts and implications of technological, social, economic and health factors that COVID-19 has on UTs and UPs teachers in terms of gender and quality of distance teaching in emergency situations, for which the following hypothesis has been established.

**Hypothesis 1 (H1).** *There is evidence of effects as a consequence of the COVID-19 pandemic on economic, academic, emotion and health of teachers of the TUs and PUs of México.*

**Hypothesis 2 (H2).** *There are differences in the affectation by the pandemic between males and females.*

The above to be in accordance with the research questions presented below.

- Question 1: How has the pandemic affected teachers during the pandemic during the distance learning process?
- Question 2: What is the main factor that affected teachers during the pandemic?

## 2. Problem Statement

The COVID-19 pandemic pushed the adoption of extreme measures to prevent the propagation of the virus. School closures radically changed the method of teaching as well as the students' and families' activities. In a few weeks, educational systems were pushed to change and adapt to the new circumstances. In Mexico, an important sector of education is composed of technological and polytechnic universities (TUs and PUs, respectively). One of the main aspects of this research was learning how teachers feel about this matter and the possible unfavorable situations of this paradigmatic change in education considering gender, age, and geographical location. Teachers were used to teaching in physical classrooms and had to become users of technological tools in order to develop distanced interactions among themselves and with the students. During this sudden change, the teachers faced personal, economic, health, and emotional pressures that impacted the development of the teaching–learning process either online or in a hybrid model. The way teachers of TUs and PUs felt highlighted the unfavorable impacts on their performance because of these personal, economic, health, and emotional pressures. Along with these impacts, the digital competencies and the consequences of the student's instruction were modified because of the 30% theory, 70% practice distribution of the educational model of these universities because there was no possibility of using the facilities for the practical component.

The present work is about the feelings and voices of the actors of HEIs from within the teaching–learning process and comparing them to the provisions and suggestions of international organizations and the national agreements due to the adaptation of the educational systems in a few weeks. In the case of Mexico, an important sector of higher education is composed of the TUs and PUs, where the main idea is to prepare students for the working environment by doing internships with different businesses, which was not possible during the pandemic. One of the main aspects of this research was learning the feelings of the teachers of the TUs and PUs to address the areas of opportunity in identified situations due to the changes in the scheme of teaching and learning. Teachers left the traditional classroom, to which they have been accustomed for decades, to become forced users of the technological and digital tools that exist to interact at a distance among themselves and their students while having to deal with the personal pressures of confinement and its economic, health, and emotional implications. These factors represent a problem in the development of the teaching process from the face-to-face to the online, and in some cases,

hybrid modality. The feelings of the teachers at the TUs and PUs in Mexico pointed out a series of impacts that were not favorable for their performance, such as the economic impact and the investment made, their digital competencies, and the consequences for the education of the students since the educational model of the TUs and PUs consists of classes with a 30% theoretical and 70% practical syllabus. Through this research with the participation of 22 TUs and PUs in 2020, we sought to analyze the impacts and obstacles presented by the students in this paradigmatic change, the competencies developed by the teachers, and the real challenges that have occurred due to the change from face-to-face to virtual classes from the teachers' points of view.

## 3. Methodology

*Research Design*

In this study, we used a non-experimental design because it did not manipulate the variables at the laboratory level, and the information was investigated at the level of observation. The required information is obtained at the field level in a single moment using a quantitative, descriptive, analytical/correlational methodology for finite populations. The variables obtained from the questionnaire and analyzed were divided into six sociodemographic categories (socio-demographics, academics, economics, emotional, social, and health). These effects were those put into practice by the participants (teachers) during the distance education modality due to the mandatory social isolation in the 2020–2021 school year. Figure 1 shows the steps developed.



**Figure 1.** Methodology.

## 4. Methodology

Figure 1 presents the methodology for this research.

- Step 1: The investigation began with the conceptualization of the exploratory study with teachers of TUs and PUs in the period of March–August 2021 to assess the impact that the paradigm shift from face-to-face teaching to remote teaching had on teachers.
- Step 2: The research design: The type of study was descriptive non–experimental transactional since data was only collected at a single point in time for the purpose of describing the academic, social, economic, emotional, and health variables.
- Step 3: Design and validation of the data collection instrument.
- Step 4: Data collection was performed. The data collection instrument was deployed in electronic media to facilitate the data collection.
- Step 5: Statistical analysis of the quantitative and qualitative results was performed.

This study was carried out with a survey of the teachers of the TUs and PUs in Mexico during the first semester of the pandemic caused by the coronavirus disease. The data was collected through an online survey tool. The results obtained from the study were divided into five categories—academic, social, economic, emotional, and health. These categories were determined by an academic commission conformed from teachers belonging to TUs and PUs summoned by the GCTPU in November 2020.

### 4.1. Data Collection Instrument

Once the objectives were established, an ad hoc questionnaire was developed that included, in addition to sociodemographic data, aspects related to the confinement caused by COVID-19. The data collection instrument of the TUs and PUs included 25 items (questions) that describe the contextual environment of each of the categories and was also.

A Likert scale with a maximum of 5 points was used for the questionnaire. A numeric value was assigned to each point to encode the categories of the downloaded responses

into a statistical system. The participants responded by choosing the point that most fit their response. The respondents were expected to respond to an assigned open question to mention the educational strategy that helped them improve their performance in the virtual classes [47]. In Table A1, the data collection instrument presents the analysis of the questions answered by the teachers of the different participating universities. The questions were classified into categories socio-demographic (Q1–Q4), academic (Q5–Q11), economic (Q12–Q14), emotional (Q15–Q19), social (Q20–Q22), and health (Q23–Q25). To validate the data collection instrument, a panel was carried out, resulting in a Cronbach's Alpha of 0.86 in the pilot test; thus, it was considered adequate. It was then delivered to 22 universities so that the participating teachers could decide to participate or not.

### 4.2. Participants

The General Directorate of Technological and Polytechnic Universities established the need to identify and evaluate the impacts of the pandemic and the confinement caused by COVID-19. In the second half of 2020, a commission of 21 teachers was formed to carry out the study and evaluation of the situation derived from COVID-19 and its effects on the educational processes. Of the total 120 technological universities and 60 polytechnics, and with the criteria of applying the data collection instrument to the institutions that had the highest number of students enrolled in the academic period of 2020–2021, the data of 15 technological universities and 7 polytechnics were used, guaranteeing the reliability of the results of the sample. Sampling a population of 12,000 teachers nationwide with a margin of error of 5% determined that 374 teachers had to answer the questionnaire [38]. For the present study, 993 responded, which reduces the error [48].

### 4.3. Multiple Correspondence Analysis (MCA)

Multiple correspondence analysis (MCA) is an extension of correspondence analysis (CA) that allows one to analyze the pattern of relationships of several categorical dependent variables. As such, it can also be seen as a generalization of principal component analysis when the variables to be analyzed are categorical instead of quantitative [49]. Because MCA has been (re)discovered many times, equivalent methods are known under several different names, such as optimal scaling, optimal or appropriate scoring, dual scaling, homogeneity analysis, scalogram analysis, and the quantification method. Technically, MCA is obtained by using a standard correspondence analysis on an indicator matrix (i.e., a matrix whose entries are 0 or 1). The percentages of explained variance need to be corrected, and the correspondence analysis interpretation of interpoint distances needs to be adapted [49].

## 5. Results

### 5.1. Analysis of the Results of the Teacher Evaluation Instrument

The teachers from the TUs and PUs are an essential part of the educational system who carry out the teaching–learning process. A total of 993 teachers of TUs and PUs at the national level were surveyed. This study allowed us to understand the real situation of teachers' lives under the conditions of the pandemic from March to August 2021.

### 5.2. General Results

The general results consist of socio-demographic data, including the participating universities (Q1), teacher education programs (Q2), gender (Q3), and the ages of the teachers (Q3).

Question 1 asked at which university the teachers worked and revealed the participating universities.

Question 2 was regarding the gender of the teachers. Male participants were the major respondents to the survey, with 55%, followed by females, with a 45% participation rate. The surveys carried out were distributed to university teachers from different careers, 4 of which correspond to administrative educational programs and 12 to various areas of engineering.

Question 3 determined the teachers' age range. A total of 37% of the teachers were in an age range of 31–40 years, followed by 28% in the age range of 41–50 years.

5.2.1. Academic Effects Results

Question 4 was "What kind of learning tools (teaching materials) do you use to teach virtual classes?" 53% of the teachers have used various digital tools to capture the attention of students. Currently, the role of the teacher is important and teachers must be creative to attain better quality in the teaching–learning process. The online teaching method was adapted to keep education going while preventing the teachers from being exposed to COVID-19 [17,18].

Question 5 was "How skillful are you in using the electronic devices to teach virtual classes?" A total of 48% of teachers responded that they use electronic devices with good and excellent skills for their virtual classes. Teachers learned to use or handle electronic devices with greater skill to teach their virtual classes. In a study conducted at two Swedish universities in 2019, the pedagogical added value was identified in their own teaching–learning context [50]. They used digital tools to increase motivation for concrete, effective, and subject-oriented successful examples, as presented by experienced teachers' performance in virtual classes.

Question 6 was "Does the speed of your internet connectivity affect your virtual classes? A total of 36% of the teachers said that the speed of their internet moderately affects their virtual classes. In addition, 22% and 26% reported that they have been affected the most by their internet connectivity. Unfortunately, most of the teachers, 48%, had difficulties with their internet signal. One of the causes for this is that they live in remote locations where the signal reception is low, causing the transmission to be lost or the image to freeze. In addition to the above, there are cases of network saturation due to several users being connected simultaneously, which translates into a decrease in speed. Therefore, internet connectivity is one of the major factors that affected the teachers' performance under these remote conditions.

Question 7 was "How adequate is the computer equipment you have for teaching virtual classes?" The results indicate that 44% of the teachers considered that they have good computer equipment suitable for teaching their virtual classes. These results are in agreement with studies conducted by Eileen Winter (2021) [51]. The author found that success in higher education depends on educators having the skills, knowledge, and competencies for online teaching. In addition, according to Juarez-Santiago (2021), the use of cell phones made the teaching–learning process more dynamic [6]. The main concern is in regard to the 11% of teachers who teach online classes through deficient equipment. This implies difficulty in showing the thematic contents, participating in activities assigned to the student individually or in groups in real time, as well as in extracurricular activities, such as homework or team meetings. This limits the participation of the teacher with the student during the class session in addition to presenting physical and mental exhaustion after a day of online classes.

Question 8 was "Do you consider that you spent an extraordinary amount of time on the preparation for virtual classes?" From the total number of respondents, 90% considered that they spent an extraordinary amount of time on class preparation under the new teaching scheme through virtual classes. The additional time was mainly due to preparing classes where laboratory practices are required, preparing slides, videos, and questionnaires, searching for information such as books, articles, software, videos, or simply literature that is available online, and helping students who lost connectivity during class for some reason or were unable to connect online to the scheduled session.

Question 9 asked the teachers if they consider that teaching virtual classes reduced their work performance? The results show that 40% of the respondents disagreed that virtual classes decreased their work performance, 23% of teachers had a neutral position, and 30% agreed that teaching virtual classes decreased their performance. This perception was manifested mainly in teachers older than 41.

Other generalized factors were the difficulty in teaching a practical topic that requires the use of a specialized laboratory, software, or equipment, and the explanation of topics that require sequential development with reactions or mathematical expressions.

Question 10 asked the teachers if simultaneously paying attention to household chores and their virtual classes affected their performance as a teacher. From the total number of respondents, 28% considered that home activities and virtual classes had a neutral effect on their performance as teachers. On the other hand, 21% mentioned that it did not affect their academic performance. According to CEPAL-UNESCO in 2020, teachers had to carry out planning adapted to the new teaching process [29]. They are now designing materials and diversifying the media, formats, and platforms used [52]. In addition, teachers had to take part in activities to safeguard the material security of the pupils and their families, such as distributing food, health products, and school materials. All this requires time and effort on the part of the teachers, and the consequence is fatigue from overwork and stress. A total of 56% considered that home activities and virtual classes in the same space affected their performance as teachers too much. One of the causes for this was that they did not have an adequate space to teach their classes online. This had a direct impact on their academic performance due to the presence of a series of distractions, such as the other occupants who were at home walking, talking, doing domestic activities, and so on [26].

Question 11 was "Do you agree that face-to-face academic programs should be complemented by online learning?" The results show that 51% of the teachers agreed that face-to-face academic programs should be complemented with online learning; 2% did not agree with this. A study conducted in Turkey to determine the effects on the education process found that teachers had both positive and negative opinions about distance education competencies, student adaptation, and communication between the teachers and parents [31]. Their results showed that the pandemic provided positive acquisitions in the technology use, awareness, and skill development in the teachers. However, they suggested that access to high-speed internet at the residences where students receive this service should be ensured. Finally, they also suggested that the provisions of these devices to students who do not have computers and laptops in their houses due to financial limitations will enable the education process to continue without any setbacks during the course of the pandemic.

### 5.2.2. Economic Effects Results

Question 12 was "How much did you spend on the purchasing of technological equipment and payments for software licenses to teach your virtual classes? The results show that 40% of the teachers spent less than MXN 5000.00 in the purchasing of technological equipment and payments for software licenses, while 31% spent up to MXN 10,000.00. Only 23% made equipment purchases above MXN 10,000.00. Joshi et al. (2021) did a study to determine the impact on teachers working in the government and private universities of Uttarakhand, India [47]. They found several institutional support barriers, such as the budget for purchasing advanced technologies, a lack of training, a lack of technical support, and a lack of clarity and direction.

Question 13 was "Has the pandemic negatively impacted your finances?" A total of 41% of the teachers agreed and 23% strongly agreed that the situation of the pandemic has had a negative impact on their economic situation. Only 3% strongly disagreed about the impact on their finances. This situation could possibly be because the teachers did not prepare for buying technology for their home office. Many institutions in Mexico provided the computers and technology to carry out the teaching activities. Teachers from India also had technical difficulties. Their difficulties were due to a lack of technical support, including a lack of technical infrastructure, limited awareness of online teaching platforms, and security concerns [30].

Question 14 asked if the situation of confinement generated expenses for services (internet, electricity, water). A total of 40% of the teachers reported that their expenses for services greatly increased during confinement, and 30% said that they had incurred

increased costs from the use of the internet, electricity, and water. The disadvantage of working at home during the pandemic is that workers can lose their motivation to work because of the electricity and internet costs [32].

### 5.2.3. Emotional Effects Results

Question 15 was "How did the pandemic affect your family interactions? The results show that 61% were not affected by the pandemic. These results are similar to those of Joshi et al. (2021) reporting that in a home office environment, a lack of basic facilities, external distractions, and family interruptions during teaching and conducting assessments were the major issues [47]. In addition, according to one report by CEPAL-UNESCO in 2020, teachers and educational staff have had to face the demands of providing socio-emotional and mental health support to students and their families, which is an aspect of their work that has become increasingly important during the pandemic [52,53].

Question 16 was "Do you think that your family requires psychological support to continue their lives during the pandemic?" The results show that 29% said they were undecided about receiving psychological support to be able to continue their lives during the pandemic. A total of 24% agreed that their family required psychological attention, while only 4% said they strongly agreed they needed psychological care. On the other hand, 35% stated that they disagreed that they needed psychological support. The teachers' personal problems, including a lack of technical knowledge, negative attitude, course integration with the technology, and a lack of motivation were identified as the fourth category to damper their engagement in online teaching and assessments [30].

Question 17 was "Do you consider that your home office invades your family's privacy?" The results show that 36% of teachers agreed and 27% strongly agreed that working online from home invaded the privacy of the family. Only 3% disagreed because they did not feel that their family's privacy was being invaded. Emami-Naeini et al. did an online study in 2021 with 220 prolific worldwide participants. They found that privacy and security were among the most frequently mentioned factors impacting participants' attitudes and comfort levels with conferencing tools and meeting locations [33]. The study revealed that most participants lacked autonomy when choosing conferencing tools or using a microphone and/or webcam in their remote meetings, which, in several cases, contradicted their personal privacy and security preferences [33].

Question 18 was "Do you agree that repetitive interaction with electronic media negatively affects your emotional state? The results show that 26% of the teachers agreed and 12% strongly agreed that repetitive interaction with electronic devices negatively affected their emotional state. However, 30% of teachers were undecided, and only 5% indicated that they strongly disagreed. Many studies have suggested that the COVID-19 pandemic has caused people around the world to exercise social distancing, and this situation has led to remote communications in working, socializing, and learning from home [35].

Question 19 was "How do you feel when you are teaching a virtual class? It was observed that 36% of the teachers were in a positive state and 25% were happy when teaching virtual classes. The psychological stress surrounding the transition to online synchronous teaching was associated with elevated levels of vocal symptoms, especially for those who reported high levels of psychological stress during previous periods of teaching. These results with teachers agreed with the notion that psychological stress may have a negative impact on the voice [36].

### 5.2.4. Social Effects Results

Question 20 was "How often do you participate in family meetings away from home during the pandemic?" The responses indicated that 93% of the teachers stated that they never or rarely participated in family gatherings outside the home. These results suggest that the teachers complied with distancing safety measures.

Question 21 was "How often do you participate in meetings with friends through virtual sessions?" The results show that 49% of the teachers rarely had meetings with friends through virtual means.

Question 22 was "Do you lead your social life through social networks (Facebook, Twitter, WhatsApp)? A total of 36% of the teachers agreed and 13% strongly agreed that they led their social lives through social networks such as Facebook, Twitter, and WhatsApp. However, 30% of the population was undecided when answering the question. The online social presence demands much more teacher–student interaction and feedback [54].

### 5.2.5. Health Effects Results

Question 23 was "How is your health condition regarding COVID-19?" The results indicate that 72% of the respondents did not have any symptoms related to COVID-19. Only 3% had a confirmed diagnosis. The confirmed cases per day ranged from 2000 to 6000 during the application of this survey [38,55] . A total of 660,185 accumulative cases were confirmed during the first semester of March to August 2021. The Mexican COVID-19 platform has a database regarding the COVID-19 cases in Mexico [38,55].

Question 24 was "Have you suffered the loss of a family member due to COVID-19?" In the first half of the pandemic, it was recorded that 19% of teachers from TUs and PUs had family loss due to COVID-19. As we can see, this result shows that at the start of this pandemic, not many teachers lost a member family. However, there were 3,846,508 confirmed cases of COVID-19 with 291,204 deaths in Mexico from 3 January 2020 to 17 November 2021 [56].

### *5.3. Statistical Analysis*

#### 5.3.1. Analysis of the Main Factors Affecting Teachers

G-test was performed to determine if there is a relationship between gender and other factors such as economic, connectivity and, training factor. Independence between factors implies that there is not difference between male and female teachers related to the contrasted factor. In this way, the G statistic obtained was 20.53 (df = 9, *p*-value $< 2 \times 10^{-16}$, the associated *p*-value was less than 0.05).

H1 is accepted: There is evidence of affectations as a consequence of the COVID-19 pandemic in the economic, academic, emotional and health of the teachers of the TUs and PUs. It was determined that the aspects that most impacted the work of the teachers in the teaching of distance classes were the computer equipment they had, the few skills in the management of information technologies and connectivity.

H2: There are differences in the affectation by the pandemic between males and females. The hypothesis is not rejected but by a statistical analysis it was determined that the main affectations were economical, the equipment and connectivity. Furthermore, although gender is commonly associated with more negative impacts for female than for male, in this study, results were the same for both. It could be because conditions were similar: training was done equally; male and female were invited to answer the instrument equally, and the percentage feeling the increase in expenses were perceived equally by male and female [41,43,45,57].

#### 5.3.2. MCA Analysis

Multiple correspondence analysis (MCA) was performed to analyze the data, and the information was processed in R software *v. 4.1.3.* The results are presented below.

In Figure 2, the dots closer to the horizontal axis are related to the following: extra expenses (Q7 and Q19), connectivity problems (Q8), computer equipment (Q9), emotional damage (Q11 and Q17), feeling more affected working from home (Q16, Q17), considered that there was more dedication and that online classes were not of the same quality as face-to-face (Q10 and Q20), presenting health issues (Q22), and losing a family member (Q23).

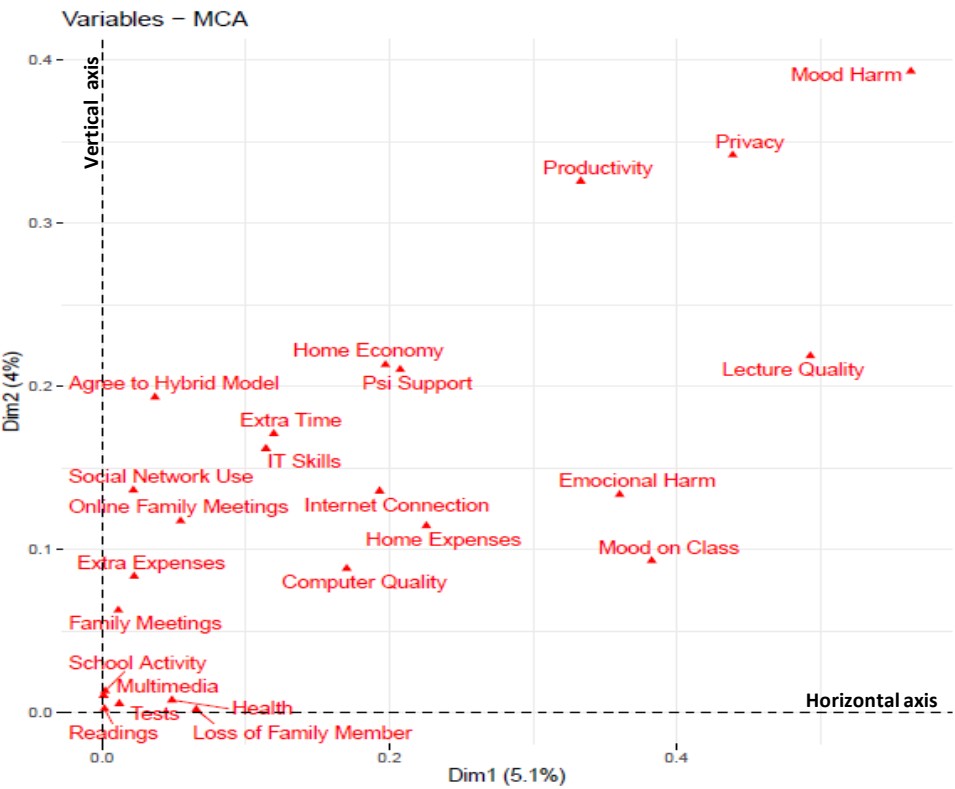

**Figure 2.** Variable Factor Map—MCA from Questions 5–25.

The vertical axis represents the teachers who stated how much they agreed with the hybrid model and the use of social networks (Q14), teachers who held online family meetings (Q12 and Q13), those who had extra expenses and equipment expenses (Q19 and Q21), and those who spent more time on the preparation of materials, such as practical lessons, multimedia activities, and readings (Q5). In axis 2 (vertical), it can be observed, according to the answers of the data collection instrument Appendix A, how much teachers agreed with the hybrid model (Q16 and Q18) and the use of social networks (Q14), how many teachers held online family meetings (Q13–Q15), the respondents who considered that distanced classes generated extra expenses for computer equipment and the internet (Q7, Q8, and Q21), and those who invested more time on the preparation of materials, such as practical lessons, multimedia activities, and lectures, and the aspects that affected their performance according to the respondents (Q5, Q10, and Q24). The horizontal axis shows that the results are divided. On one side are the teachers who did not feel comfortable, did not agree with the hybrid model, and were more accustomed to face-to-face meetings during the pandemic but felt that they had few digital skills. On the left side, we can observe the number of teachers who were comfortable with the online and hybrid model (Q18), did not have many expenses, did not invest a lot of time in doing extra activities, and felt that they had very adequate digital skills to carry out the teaching–learning process (Q6). These teachers in turn invested their own resources in their training to teach online (Q5, Q6, and Q25). They also considered the need to complement face-to-face academic programs with online models.

Figure 3 shows the results corresponding to Question (Q18). On the left side of the graph are the teachers in favor or who felt comfortable with the distance learning scheme; on the right side are teachers who did not feel comfortable with the distance learning model. This was equivalent to the teachers who were not comfortable with online classes (right side). According to the data, 50% of the teachers felt uncomfortable with the distance learning model and 50% of the teachers felt comfortable with the distance learning model.

The value of 154 in quadrant I corresponds to the outliers since it corresponds to the teachers who showed a high degree of commitment to their work and who felt comfortable with the distance learning model, produced materials of different types, and even invested in their training. In quadrants I and II, the results correspond to the teachers with high commitment and great dedication who developed a wide range of work materials to maintain the quality of the teaching–learning process. In quadrants III and IV, there is a concentration of teachers who limited themselves to providing few activities and materials to teach their classes.

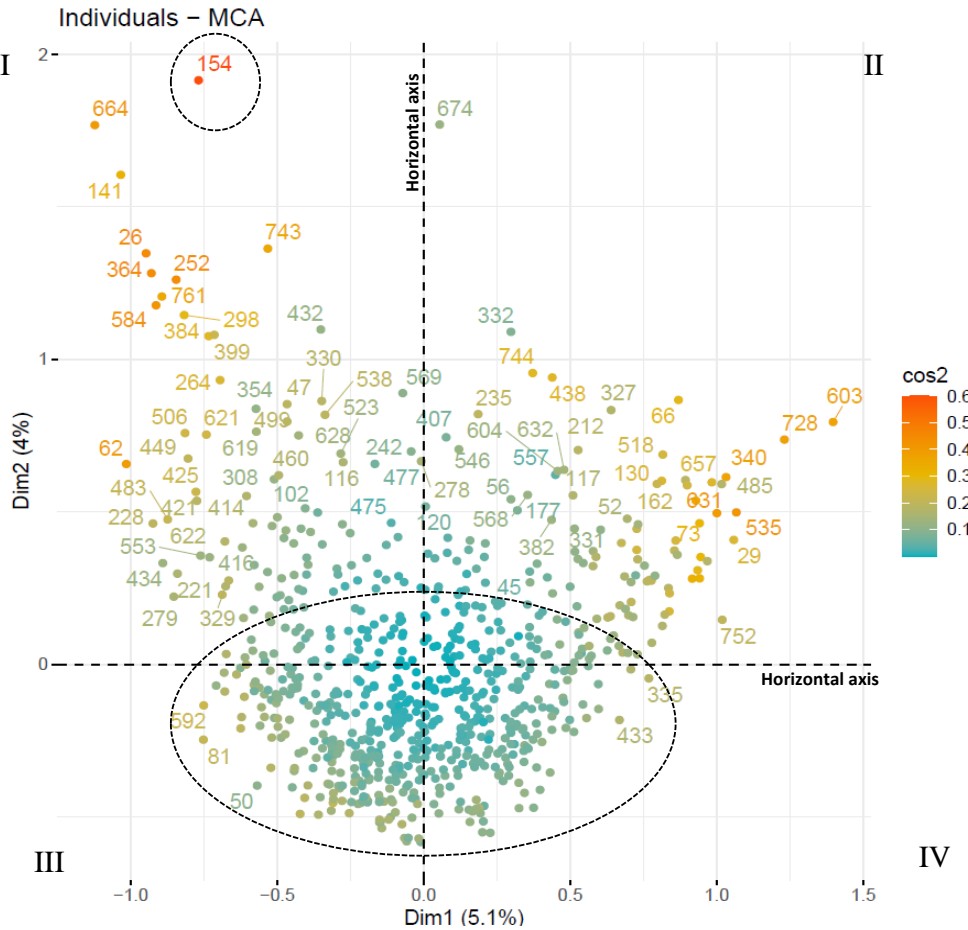

**Figure 3.** Individual Factor Map—MCA from Questions 5–25.

Figure 4a shows the results of the analysis by university. Figure 4b presents the universities. The upper quadrants (I, II) show the universities that had the fewest problems in dealing with the pandemic in relation to technological infrastructure, connectivity, and the willingness of students to connect Figure 4a. This group corresponds to the universities located in the central and northern regions of Mexico Figure 4b.

The lower quadrants (III, IV) show the cluster of universities that presented problems with their technological infrastructure and connectivity and had more cases in which students did not connect to classes or experienced lags. This is understandable because most of these institutions are located in socioeconomic areas of lower socioeconomic development in Mexico in the states of the south-southeast region of the country.

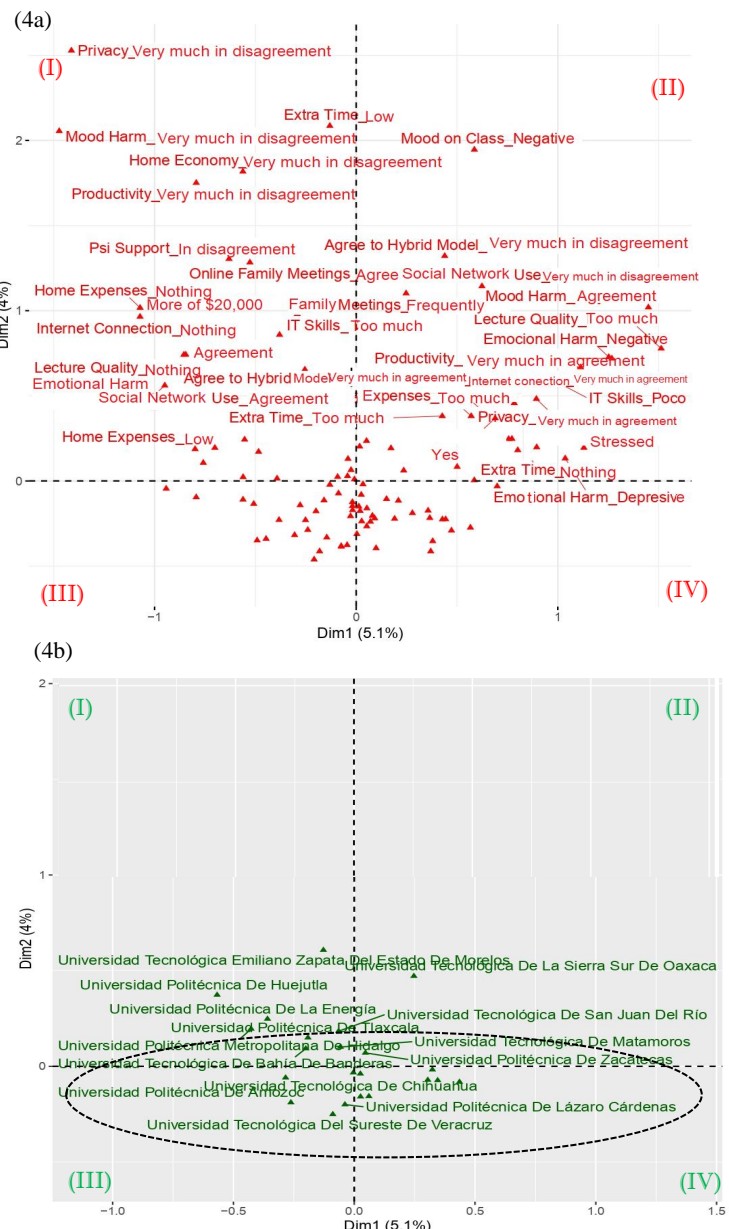

**Figure 4.** (**a**) Biplot Factor Map—MCA (**b**) Universities from Questions 5–25.

## 6. Discussion

Precautions to protect the teachers, students, and community from the possible spread of COVID-19 in school environments were essential. Citizens thought that the health emergency would not last for long and did not understand the reality that students and teachers would face [58].

In regard to the effects of the COVID-19 pandemic, the findings from this work show that the teachers in TUs and PUs in Mexico demanded a high intellectual level. Different studies have pointed out that the pandemic caused an abrupt change without a previous announcement. Therefore, not being prepared led to some difficulties and shortcomings, and teachers were not free from the consequences. Hence, many countries experienced high workloads during the health emergency crisis [58]. Worldwide, HEIs transformed their teaching–learning process into "online teaching–learning", which is generally considered a distance learning method through the internet [51]. When COVID-19 forced people to stay at home, online learning developed an important role in helping students continue their

studies through virtual lessons, allowing them to attend from anywhere in the world. In India, it was suggested that online classes could be as effective as the face-to-face ones if provided with the necessary tools. Moreover, online learning has been essential in reducing the mental stress and tedium of the lockdowns [52]. Other authors have also pointed out that spending too much time being inactive could affect one's health and wellbeing, such as the case with depression and anxiety [53,59]. Other studies have indicated the limitations of online teaching and learning, including the deficiency of the online teaching infrastructure, the limited exposure of teachers to online teaching, the information gap, family houses being unfavorable environments in which to learn, and equity and academic excellence in terms of high education. Authors have also suggested the need to adjust different pedagogical approaches to different content and subjects. Although the research literature in this field is progressing, various authors have indicated the limitations and strengths of online learning compared to the face-to-face modality [54]. Furthermore, the challenges of online learning are relevant among and within countries. The COVID-19 lockdowns depleted the scarce resources for disadvantaged students because there was no chance to visit libraries or other public spaces to get an internet connection or a space to study [53,59]. HEIs were the first to replace their face-to-face lectures with online learning when the lockdown started to prevent the spread of the virus [17]. They transformed their learning environment to open the scope of digitalization, complementing the student–teacher relationship [1,58]. However, pedagogical transformation is not easy for teachers practicing traditional teaching methods, and they need adequate training to improve their pedagogical knowledge, skills, and competencies [60,61]. Training could help them manage teaching in an efficient way and, therefore, meet the learning needs of the students. This is why it is important to inquire about the way in which the education systems of countries with firm traditional learning methods are handling the pandemic. The abrupt and contagious advancement of this disease provided limited time to prepare and plan, which makes the question even more interesting [21,51]. In previous studies, it has been shown that the development of the digital literacy of teachers has become vital. Research regarding the increases in teaching skills and awareness of the necessary conditions for digital literacy is a hot topic. Digital literacy is a potential competency in the professional development of teachers in academic learning communities for a global economy [62]. Digital literacy in teachers is referred to commonly as the necessary skills and knowledge teachers need to learn in the digital knowledge society [8]. Recent studies have suggested that improving the digital literacy of teachers profoundly changed the conventional teaching–learning model. However, with the progression of digital technology, some risks of digital inequities seem possible. Many factors could have an impact on teachers. Some teachers considered that the training in digital literacy was arduous and required too much time. They felt challenged to develop digital competency within the existing professional development plans, which is why the acceptance of digital training was reduced. To a certain extent, the development of the digital co-dependence of the teachers varied with the preparation of the academic personnel [17], and their readiness relied on the flexibility of the school platform [6]. The effective use of digital technology in schools requires deep digital literacy changes in the teachers. In our current situation, digital literacy skills are essential for education in online networks and communities. In the long term, digital tools will be integrated with conventional teaching, remodeling the digital literacy of teachers in a sustainable way [51]. Online lessons have difficulties and problems. For example, both teachers and students do not have much experience teaching and learning virtually. Furthermore, not all families have internet access, and, in many cases, students do not own a computer or digital device [15]. These factors could generate stress and demotivate people, increasing the economic problems of families with unemployment, the loss of housing, and the digital gap. This is important because it should be considered in the public policies to economically, educatively, and psychologically help those less advantaged in the best possible way. From the obtained results, and after analyzing registered data, it can be stated that most of the teachers would prefer confinement to end [19]. The results of the study are not very different from other studies published in different media. Each teacher has had to use their own means, effort, knowledge, and existing competencies to facilitate a service not previously

experienced, and they have been guided, in our opinion, by an awareness of the seriousness and a commitment to solidarity during the pandemic. The results show a generally high level of dedication, although the measures of virtual and online universities and educational innovation should consider if these are the best practices for these channels. The COVID-19 context has required educational institutions to change their management styles and adapt to digital technology to address students. Education needs practical and direct interaction to achieve the learning goals by getting used to educational platforms, such as MOOC, VR/AR, games, heart rate monitors (HRMs), CoachMyVideo, MyOMaps, GoNoodle, and so forth [63]. Using technology and a change in the strategy of classic management to agile management by educational institutions could help improve the SPE results, facilitating teaching, development, and learning.

A great majority of teachers that faced the rapid change from face-to-face to online teaching fought the adaptation to tension and additional workloads. As a consequence, the professional role of teachers changed, diminishing their satisfaction levels and bringing lots of digital literacy challenges. They had to balance their responsibilities, teaching, labor, and the rush to change to online teaching urgently. Teachers had obligations that also led to psychological pressure due to the lack of technical support. There are still online teachers from different academic levels, ages, and backgrounds with inadequate or no digital literacy preparation.

In the present study, the role of the teachers and their satisfaction during the pandemic was investigated. At the same time, we researched the development of digital literacy in teachers and how it helps to interpret the role of teachers with a wider scope to increase their satisfaction. These intertwined aspects could increase the quality of education [64].

Nikolopoulou (2022) studied the positive and negative aspects of the lockdowns in Greece. His findings showed that most of the teachers used a combination of teaching practices and scopes, while the implemented learning activities were mainly of language, psychomotor education, presenting videos, math and interdisciplinary activities. Teacher's feelings were negative initially, but changed to more positive afterwards. The disadvantages of online education, as experienced by the teachers, were mostly related to technical problems, followed by limited resources/support for students at home, and limited training in online methodology. The positive experiences of the teachers included increasing the knowledge students have about technology and maintaining the academic environment [57]. Such findings match with the results obtained in this study, where in the final step, teachers felt comfortable with the online model.

For a sustainable educational method, online technology should be an option, but it should not be considered permanent for accomplishing academic and labor goals [65]. It should be a flexible method of learning during emergency situations, such as COVID-19, as it contributes to developing skills and knowledge with a different delivery, but strict and prudent measures are required to ensure the benefits. Academic achievements are an efficiency measurement of a sustainable education approach. There are no reciprocal connections between academic success and preparation for labor, and somehow, academic achievements were overestimated during the pandemic. An education method based on online technology for academic success and preparation for work would become dysfunctional. In general, we should explore a comprehensive approach to sustainable education via online technology during the pandemic.

The present study found that the professional roles of the teachers changed in a complicated manner and that the workload for teachers increased during online teaching. With the change in the educational environment due to COVID-19, it has become urgent to improve digital literacy. From the perspective of educational administrators, more can be done following the correlation of the three categories mentioned in this study due to the difficulties and limitations teachers faced. Focusing on real changes in the status of online higher education as an agent of change for sustainable education and preparation for work could provide a new perspective to benefit society. Meanwhile, the perceptions of both students and teachers could be improved by using online education as a way to improve the declining economy or by reducing costs for online training or education.

Similar to other HEIs, Mexican universities adopted online learning during COVID-19. Considering the face-to-face and traditional teaching background, transforming the process to online was difficult. Moreover, the rapid change did not give the teachers time for the pedagogical use of information and communication technology (ICT), and most of them used their knowledge and skills to learn how to create videos and manage classes independently, while some others depended on external support, such as from their family. This study revealed that at the beginning of the pandemic, teachers had a negative perception of online learning, which in turn impacted the confidence and motivation of both the students and teachers. However, both gradually recognized the benefits that it could offer. This finding adds relevant information and is similar to the results of other studies conducted worldwide [66,67].

Online learning could be more effective if used with pertinent didactics and approaches. Despite the challenges, the COVID-19 pandemic could achieve the scope of the transformation of traditional teaching into an interactive method that is centered around the student. In this study, it was found that online learning has limitations related to the lack of social interaction and practical lessons, although, the teaching–learning perspective has its benefits in the short and long term. It helps protect against the anxiety and depression that might occur with isolation and a lack of activities. Online learning could transform traditional teaching into a student-centered process that integrates ICT and reinforces teachers' capabilities [51].

Special mention should be considered to Malish et al. [45] study about the gender related aspects during the pandemic. Although it is a global issue and education makes no exception, the mental, physical, social and economic impacts attributed to lockdowns are amplified by pre-existent inequities that should had been born during the pandemic.

The present work provides an initial view regarding the first steps of the transition to teaching using digital devices in April 2021. The results reflect an educational community in great uncertainty regarding the process and the future of education due to the period of 2019–2020 and the confinement situation in March 2021. This perception is not sustainable and is negative for the student community, which, at the time, was very concerned about their academic preparation. One result that stands out is the workload of both teachers and students; both perceived an increase of 9 h per week for class preparation and e-learning. This result has statistical significance and is surprisingly similar in both cases. In terms of evaluation, the concern and uncertainty were evident during the survey period. Furthermore, there was a lack of guarantee in the whole process and a high percentage of the respondents indicated that the resources were not adequate. Regarding the difficulties in obtaining class credits, the students did not agree with the teachers. The students said that it was more difficult to pass the subject (83%) versus 37% of teachers. According to the data, the commitment to distance learning was low during the survey period. A total of 43.2% of teachers would not mind adapting to e-learning, although only 32.9% of students preferred it. This could be because of the traumatic transition to virtualization because of the forced lockdowns. In conclusion, the data from this study reveal that emergency remote teaching (ERT) requires a methodology for the better use of tools to reinforce the required quality education. More research and participation are required from the education managers to offer solutions to the identified problems and develop a sustainable transition to different conditions. The main problems identified were a lack of communication and implementing a fair and adequate grading system that reduces the digital gap. These problems have been pointed out in this research.

Class suspension matched with the first vacation period of the year and there were no academic activities for at least two weeks. Once this period ended, SEP decided to continue classes remotely. The decision took students and teachers by surprise and faced the challenge of virtual education. Soon they experienced the drawbacks of a system that was too dependent of the face-to-face modality and from the technological access. The relations between teachers and students was mainly by internet in virtual classrooms, email, SM and also the retake of the activities, with academic communities with no means to take it over, such as connectivity, lack of resources and training [46].

Many of the teachers, both male and female, were not qualified to such transition. The lack of training had many causes: not all homes were prepared for the technical requirements related to TIC, and not always is possible to maintain a comfortable environment for home office. Sometimes the work load requirements, due to adaptation difficulties, isolation, and social distancing increase. Being at home, teachers should manage many factors: people in a small place, children, spouse and people working and studying in the same place. This reality could be different for male and female, standing out the gender differences that affect family relationships and also the labor market. Although the literature focus in female exposure to domestic violence due to the quarantine, the work environment legitimately considers female as overwhelmed, without ruling out the affectations male face [46].

*Limitations*

This study has limitations that were identified by the authors. One of the limitations was confinement. The results were processed from surveys that were answered by teachers voluntarily during the period between March and August 2020, with a non-probabilistic self-selected sampling. The analysis population consisted of TU and PU teachers, with a total of 993 teachers, which did not necessarily give the same results as other HEIs in Mexico. Due to the selection of the participating teachers, the generalization of the results should be carefully analyzed. These limitations provide a clear path for future research, such as the construction and validation of a more in-depth and robust instrument to learn about the perceived differences between working or studying online and face-to-face interactions in order to replicate the research with a larger sample of higher education teachers.

## 7. Conclusions

During this work, the established objectives were accomplished, which included identifying the sociodemographic aspects of the teachers that have had an impact on the use of virtual learning platforms (VLPs). In addition, we found that the main aspect affecting teachers was the geographical locations of the universities.

Another objective was to understand the perceptions of the teaching staff of the higher education institutions (HEIs)—both technological and polytechnical universities (TUs and PUs)—regarding the use of VLPs during the COVID-19 crisis. We were able to identify, through the research, that half of the surveyed teachers felt comfortable with distance learning, while the other half felt uncomfortable. We also aimed to identify the main problems teachers faced in the teaching–learning virtualization and how they might contribute to the planning strategies of HEIs for this change. Once the problems were identified, it was shown that the main one was confinement because it impacted the educational model of 30% theoretical and 70% practical lessons. The other issues included the problems of connectivity, information infrastructure, and the digital literacy of teachers.

*7.1. Response to the Hypothesis*

This work provided an answer to the hypothesis (H1): There is evidence of effects as a consequence of the COVID-19 pandemic on the economy, academia, emotion and health. It was identified that the lack of adequate computer equipment, connectivity and digital skills were the aspects that most negatively affected the mood and productivity of teachers in distance modality. It even affected more than the loss of family member. The effects on the teaching-learning process that can be seen in the map of factors in the distribution of universities hit the most in the south-southeast region and mountain areas. Universities in those areas faced the lack of IT infrastructure, connectivity problems and inadequate digital skills of professors, differing from universities in the central-northern areas. Hypothesis (H2): There are differences in the impact of the pandemic between male and female. According to the results of the G-test, this hypothesis determined that 50% of the professors feel comfortable with the distance education model, with no differentiating element between male and female or age. The Hypothesis is not rejected but due to results and statistical analysis it was determined that the main affectations were economical, the equipment and connectivity. Although gender is commonly associated with more negative impacts than male, in this study, the conditions

were the same and results were the same for both. Training was done equally, both, male and female were invited to answer the instrument equally, and the percentage feeling the increase in expenses were perceived equally by male and female.

### 7.2. Answer to the Research Questions

Question 1: How did the pandemic affect teachers during the pandemic during the distance learning process? The geographic location of the university stands out regarding the effects of the pandemic. Teachers at universities in north central Mexico had different considerations than the teachers at universities in the south-southeastern region and mountain areas. In the latter group, there were pre-existing connectivity, infrastructure, and teacher digital literacy problems independently from the pandemic.

Question 2: What is the main factor that affected teachers during the pandemic? The MCA analysis proved that the main affectation factors were digital skills and the ability to use digital devices and VLPs for online teaching and the extra expenses and investments into technological equipment because many teachers at the universities in mountain zones had to share their digital devices with their children. Another problem was connectivity and the extra time invested in planning and preparing material for classes. In regard to health, we can mention the emotional effects of the pandemic on the interactions of families, physical health, and the loss of family privacy. Economically, the main problem was due to the expenses of computer equipment and software licenses. The main findings are described below for each category.

### 7.3. Academic Effects

The teachers indicated that they had adequate equipment to teach their classes. However, their internet connectivity was not adequate for carrying out their academic activities. Likewise, teachers stated that working from home invaded their family privacy and their work performance. Based on the above, the teachers stated that they invested a lot of extra time into preparing classes under the new scheme. Virtual classes have decreased their work performance since, as previously stated, more time is required to prepare the classes. On the other hand, teachers skillfully handle electronic devices when teaching classes and this helps them provide quality lessons. The teachers use didactic material such as exams, presentations, and practical lessons according to the thematic contents in person to provide better education to the students. Finally, they suggested that such materials should be complemented with online learning.

### 7.4. Economic Effects

The teachers reported that the COVID-19 pandemic has had a negative impact on their finances, resulting in higher expenses due to the payments for services and the acquisition of computer equipment and software licenses for teaching at home.

### 7.5. Emotional Effects

The teachers stated that their moods were modified while teaching their virtual classes, causing stress, happiness, or negativity. However, most remained in a positive state and commented that they did not require psychological support to continue working in confinement. Interaction with electronic devices did not negatively affect many personally nor did it affect their families.

### 7.6. Social Effects

The teachers considered that there were no negative impacts on their family life. The trend is that academic staff have had reduced face-to-face meetings with family and friends, and have been forced to use social networks such as Facebook, Twitter, and WhatsApp to keep in touch.

### 7.7. Health Effects

The health conditions of the teachers are good in the face of COVID-19 since most have not been infected. However, 19% of teachers have suffered the loss of a family member during this pandemic.

*7.8. Final Conclusions*

It can be concluded that the COVID-19 pandemic has been a unique, uncontrollable, irreversible event that we have had to live through, and which we were not prepared to face. However, at the national level, every sector has made great effort to cope with it, especially the educational sector at all levels, which has been linked to making major changes by moving from face-to-face teaching to online (virtual) platforms. This change has had positive and negative impacts on the parties involved. The changes in the teaching strategies are challenges for all educational sectors and they will be substantive aspects in the present and future. The use of technology for the development of classes, a lack of knowledge of virtual teaching platforms, inadequate or failure of internet connectivity, a lack of student collaboration, high pressure in the work environment, and an excessive load of tasks or activities are some of the aspects that will continue to be addressed as a part of everyday life. This study contributes to establish improvement strategies GCTPU considering online teaching and to the teachers professional development, independently from their gender. This study applies specifically to the PUs and TUs which had no previous studies for emergency situations.

## 8. Future Works

Supported by the results of this research, it is essential to establish new research branches focused in the analysis and implementation of the best practices. Wellbeing, mental health and gender equity should be developed, being the latter part of the political discourse in Mexico and the world and should be deeply studied in teachers of technical education.

**Author Contributions:** Conceptualization, L.V.-L. and R.G.-G.; methodology, L.V.-L., M.A.Z.-A., H.R.R., M.A.C.-P. and J.R.R.; formal analysis, M.A.Z.-A., W.J.P.-G., M.A.C.-P. and A.R.Á.-S.; investigation, L.V.-L. and T.O.-E.; resources, T.O.-E. and H.R.R.; data curation, W.J.P.-G., M.A.C.-P. and M.A.Z.-A.; writing—original draft preparation, T.O.-E. and L.V.-L.; writing—review and editing, M.A.Z.-A., J.R.-R., M.A.C.-P. and R.G.-G.; supervision, J.R.-R. and R.G.-G. All authors have read and agreed to the published version of the manuscript.

**Funding:** This research did not receive funding.

**Institutional Review Board Statement:** The study was conducted according to the guidelines of the Declaration of Helsinki and submitted at to participating universities for approval and distribution. The Project was approved in the Ethics Committee session. Date 2 September 2020.

**Informed Consent Statement:** Respondents consent was waived due to its minimal risk to subjects that will not adversely affect their rights and welfare. It was obtained by voluntarily answering and informing that it was for academic and statistical purposes only.

**Data Availability Statement:** Data are available on request to Lourdes Vital López.

**Acknowledgments:** The authors would like to thank the professors who responded to the data collection instrument and the participating universities: Universidad Tecnológica de la Selva, Universidad Tecnológica de Chihuahua, Universidad Tecnológica de La Costa Grande De Guerrero, Universidad Tecnológica de la Región Norte de Guerrero, Universidad Politécnica Metropolitana de Hidalgo, Universidad Politécnica de Huejutla, Universidad Politécnica de la Energía, Universidad Politécnica de Lázaro Cárdenas, Universidad Tecnológica Emiliano Zapata del Estado de Morelos, Universidad Tecnológica de Bahía De Banderas, Universidad Tecnológica de la Sierra Sur De Oaxaca, Universidad Tecnológica de Tecamachalco, Universidad Tecnológica de Tehuacán, Universidad Politécnica de Amozoc, Universidad Tecnológica de San Juan Del Río, Universidad Politécnica del Golfo de México, Universidad Tecnológica de Tamaulipas Norte, Universidad Tecnológica de Matamoros, Universidad Tecnológica de Nuevo Laredo, Universidad Tecnológica de Tlaxcala, Universidad Politécnica de Tlaxcala, Universidad Tecnológica del Sureste de Veracruz, Universidad Politécnica de Zacatecas. The authors would like to thank the members of the following Academic Groups: the Academic Group of Chemistry of Universidad Tecnológica de San Juan del Río, the Academic Group of the Maintenance of Universidad Tecnológica de Tamaulipas Norte and the Academic Group of Software Development of the Universidad Tecnológica de la Region Norte de Guerrero.

**Conflicts of Interest:** The population of this study are professors from TUs and PUs belonging to the General Coordinator of Technological and Polytechnic Universities (CGUTP). The data collection instrument was previously designed and then sent to the professors of each participating university with a Google Forms link. The response to the questionnaire was voluntary which made it a convenience sampling method. The survey header had a note informing about the use of the data to be collected and the voluntary response to provide answers as a consent. The data will be analyzed and shared only within the research team, with identifiers destroyed. The nature of the data is electronic and does not include biological specimens or sensitive information. Only the research team had access to the data and identifiers were destroyed (anonymized data). The data were stored on a computer managed solely by the research team and password protected. The data files were managed by the research team with the identifiers already destroyed and will be kept in a file with the final version of the article as evidence of the research for institutional purposes. Respondents accessed the data collection instrument through a link sent by the Research Group, with the collected data addressed to the research team. It was anonymous and voluntary response with minimal risk to the participants not exposing them to psychological, social, or physical risks, belonging to the definition of research for generalized knowledge, which makes the survey of this study exempt and was reviewed and authorized by the Ethics Committee of each participating university.

## Appendix A

Dear Professor, currently a research group composed of teachers from 22 TUs and UPs is developing an analysis of the effects caused during the pandemic and the impacts on the performance in the teaching-learning process.

The purpose of this questionnaire is to analyze the impact of the pandemic on teachers at UTs and UPs and to obtain information about the possible effects on the teaching-learning process. Personal data are protected under Mexican law and will not be disclosed. By participating in this survey, data will be shared with each participating university so that the main challenges and opportunities for the future can be analyzed and identified.

**Table A1.** Data Collection instrument.

| Questions | | |
|---|---|---|
| Q1. Write your affiliation university | | |
| Q2. Name of the career where you teach: | | |
| Q3. Gender | Male | Female |
| Q4. Age Range | 20–30 years | |
| | 31–40 years | |
| | 41–50 years | |
| | 51–60 years | |
| | 61 onwards | |
| Q5. Select the learning objects (didactic material) that you have produced to teach virtual classes | Practices | |
| | Exams | |
| | Presentations | |
| | Videos | |
| | All of the above | |
| | None of the above | |
| | Others | |
| Q6. Skillfully handles electronic devices to teach virtual classes | Too | |
| | Much | |
| | So so | |
| | Little bit | |
| | Nothing | |

**Table A1.** *Cont.*

| Questions | |
|---|---|
| Q7. The investment in the purchase of technological equipment and software license payment to teach your virtual classes is | Less than $5000.00 |
| | From $5000.00 to $10,000.00 |
| | Plus $10,000.00 to $15,000.00 |
| | More than $15,000.00 to $20,000.00 |
| | More than $20,000.00 |
| Q8. The speed of your internet connectivity affects your performance when teaching your virtual classes | Too much |
| | A lot |
| | So so |
| | Little |
| | Nothing |
| Q9. The computer equipment you have for teaching virtual classes is | Very suitable |
| | Suitable |
| | more or less suitable |
| | Inappropriate |
| | Not at all adequate |
| Q10. The extraordinary time spent preparing classes under the new teaching scheme through virtual classes is | Too |
| | Much |
| | So so |
| | Little bit |
| | Nothing |
| Q11. The emotional effects of the pandemic on family interaction have been | Positive |
| | Happy |
| | Positive and negative |
| | Depressant |
| | Negative |
| Q12. The frequency with which you hold meetings with family away from home during the pandemic is | Very frequent |
| | Frequent |
| | Regularly frequent |
| | Infrequent |
| | Not at all frequent |
| Q13. Frequency with which you participate in meetings between friends through virtual means is | Very frequent |
| | Frequent |
| | Regularly frequent |
| | Infrequent |
| | Not at all frequent |
| Q14. His social life is carried out through social networks (Facebook, Twitter, WhatsApp) | Strongly agree |
| | In agreement |
| | Neither agree nor disagree |
| | In disagreement |
| | Strongly disagree |
| Q15. Considers that his family requires psychological support to continue his life during the pandemic | Strongly agree |
| | In agreement |
| | Neither agree nor disagree |
| | In disagreement |
| | Strongly disagree |
| Q16. Working from home online invades family privacy | Strongly agree |
| | In agreement |
| | Neither agree nor disagree |
| | In disagreement |
| | Strongly disagree |
| Q17. Repetitive interaction with electronic media negatively affects your emotional state | Strongly agree |
| | In agreement |
| | Neither agree nor disagree |
| | In disagreement |
| | Strongly disagree |

**Table A1.** *Cont.*

| Questions | | |
|---|---|---|
| Q18. When teaching my virtual classes I feel | Positive | |
| | Happy | |
| | Positive and negative | |
| | Depressant | |
| | Negative | |
| Q19. The situation of the pandemic has generated a negative impact on its economy | Strongly agree | |
| | In agreement | |
| | Neither agree nor disagree | |
| | In disagreement | |
| | Strongly disagree | |
| Q20. Considers that teaching virtual classes decreases their work performance | Strongly agree | |
| | In agreement | |
| | Neither agree nor disagree | |
| | In disagreement | |
| | Strongly disagree | |
| Q21. The pandemic has generated higher expenses in the payment of services (Internet, electricity, water) | Too | |
| | Much | |
| | Regular | |
| | Little bit | |
| | Nothing | |
| Q22. Your health condition regarding COVID-19 is | Confirmed diagnosis | |
| | Suspect diagnosis | |
| | Negative diagnosis | |
| | Related symptoms | |
| | None of the above | |
| Q23. Has suffered the loss of a family member as a result of COVID-19 | Yes | No |
| Q24. Simultaneous attention to household chores and their virtual classes affect their performance as a teacher | Too | |
| | Much | |
| | Regular | |
| | Little bit | |
| | Nothing | |
| Q25. Agree that face-to-face academic programs should be complemented by online learning | Strongly agree | |
| | In agreement | |
| | Neither agree nor disagree | |
| | In disagreement | |
| | Strongly disagree | |
| Q26. Mention an educational strategy that helps you improve performance in your virtual classes | | |

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
