# Peer review of "The Impacts of COVID-19 on Technological and Polytechnic University Teachers"

_sustainability, doi:10.3390/su14084593_

Round 1

Reviewer 1 Report

Dear author(s),

Thank you for the opportunity to read your work. Please, see below some suggestions for your evaluation if they can contribute to improve your work.

The article is not in the usual format (indicating the number of lines in the text), so I used the sections to indicate the suggestions/questions

Abstract

Please explain: What is MCA?

Consider including synonyms such as pandemic and faculty in keywords, so your study can be found if published

Introduction

Missing space in the words “effectsof”

There is a full blank page (page 2 of the pdf)

There are several mentions of "distance--learning", “online--teaching” and "teaching--learning " with two dashes

The format of in-text citation is incorrect (e.g., Melaku in 2021, Gregori in 2021, etc.)

Missing space in “mong Arsi”

The following sentence needs clarification. It is not possible to understand the relationship and relevance to the introduction of the study: “There are also, some studies verifying the effects of myofascial release effects (MFR) in teachers’ posture, muscular tension, and voice quality. They found that MFR had an immediately effect in improvement of the posture, especially related with head”

The following statement require reference(s): “Other requirements for teachers include voice modulation and getting used to the techniques of the equipment and software that will be used to optimize its use, taking care of body language, and use a script to optimize the use of time, allowing the students to express themselves freely”

The following sentence needs clarification: “knowledge transfer by teachers and found that most of the knowledge transfer by teachers”

Incorrect spelling: "stu-dents", “higher–education institutions”, “techno- logical”

Please explain the differences between the types of higher education institutions, so that this information adds value to the text and to the reader;

Why did you choose TU and PU? What are the implications of this choice?

Please explain how the following objectives are connected to the research questions: Identify the socio–demographic aspects of the teachers that has an impact on the use of virtual learning platforms (VLPs); to know the perception of teaching staff on the higher education institutions (HEIs), both technological and polytechnical universities (TUs and PUs) over the use of VLP during the COVID-19 crisis;

Problem statement

The first research question needs to be much more specific (Question 1: How the pandemic affected teachers?)

Related to “Question 2: Are there differences in the affectation by the pandemic between male and female?” - Are there no studies that answer this question? How is your study different? Why does this question not appear in the research objectives?

Related to “This work intended to identify the voices and feelings of the inter- nal stakeholders of higher education,” – This is different of what you stated as your research objectives

Correct: “mexican case”

Correct: “TUs an PUs”

The following sentences need clarification:

“performance with these said impacts (pressures)”

“students instruction were modified because of the 30% theory, 70% practice distribution of Educational Model of these universities”

“Through the research, applied in 22 TUs and PUs during 2020, we were looking to analyze the impacts and obstacles teachers had in this paradigmatic change, as in the competencies developed by teachers, real challenges in the change from face-to-face to virtual classes, and the voice of the teachers.” – This is very confusing. Is it another research? Was it published? How this add to the text?

“an important sector of higher education is integrated by the TUs and PUs” – How is this important? There is a need to explain to the reader, mainly for those that do not live in Mexico

The following sentence is confusing: “In this sense and the possible unfavorable situations of the change of scheme in teaching and learning”

Consider the following paragraph: “The feeling of the professors of the TUs and PUs in Mexico, pointed out a series of impacts that were not favorable for their performance such as the economic impact and the investment made, their digital competences and the consequences in the formation of the students since the Educational Model of the TUs and PUs implies classes with 30% of theory and 70% of practice” – Where these data came from? There are repeated sentences here and in the Introduction

Missing space: “pointof”

Methodology

In the explanation of step 2 there are two colons in a row

The figure must be self-explanatory; it is not possible to understand step 4, it is necessary to indicate in the figure that it is about data collection so as not to require the reader to make assumptions

The following sentence is confusing: “This study was carried out in teachers”

Consider the following: “The result obtained from the study was divided into five categories: academic, social, economic, emotional and health.” – Please explain why did you choose these categories and not others.

Consider the following: “The student measuring instrument of the TUs and PUs includes 25 items (questions) that describe the contextual environment of each of the categories” – Please explain how did you arrive in these 25 items? What were the methodological approaches adopted to ensure reliability that such questions describe what they should?

Consider the following: “The sample is not representative, but all subjects (teachers) in the population had the same probability of being part of the sample” – This needs more explanation. If this sample is not representative, how representative is it?

Consider the following: “Technological University of Tecamachalco has the highest participation with 26.9%, followed by the Technological University of Tamaulipas North with 10.5% and in third place is the Polytechnic University of Amozoc which has 7% representation.” – Please explain how this information add to the knowledge and how it contribute to answer the research questions and objectives

This sentence is repeated several times throughout the text: “The study of the social, economic, academic, emotional and health effects caused by COVID-19 in students is focused on 15 TUs and 7 PUs”

These sentences need to be rewritten: “Question 2, gender. Male...”, “Question 3, teachers age range. 37% of the teachers”, “In Regards to question 13”, “Question 19, how do you feel when you are teaching virtual class? it is observed that”, many others – Please verify the entire results section for correcting spelling and punctuation

In the previous sections you use the expression "teacher" and in the results you use "professor". This needs clarification as they have different meanings according to countries. The text needs to be clear so that the reader can understand what is being said.

There are parts of the text that are not formatted accordingly

There are words in red in the discussion

Author Response

Reviewer 1

Abstract

Please explain: What is MCA?

Thank you very much for your comments, the correction was made in the abstract. See abstract line 13. Added section 4.3 see lines 324-335.

Introduction

Missing space in the words “effectsof” PAGE 1 There is a full blank page (page 2 of the pdf) There are several mentions of "distance--learning", “online--teaching” and "teaching--learning " with two dashes            PAGE 1

The format of in-text citation is incorrect (e.g., Melaku in 2021, Gregori in 2021, etc.)

Missing space in “mong Arsi”

R=Thank you very much for your comments, the correction was made. See Introduction  and adjustments have been made to the citation in the document.

The following sentence needs clarification. It is not possible to understand the relationship and relevance to the introduction of the study: “There are also, some studies verifying the effects of myofascial release effects (MFR) in teachers’ posture, muscular tension, and voice quality. They found that MFR had an immediately effect in improvement of the posture, especially related with head”

R= Thank you very much for your comments, the modifications have been made in the document. The sentence was eliminated as it was considered relevant to the work.

The following statement require reference(s): “Other requirements for teachers include voice modulation and getting used to the techniques of the equipment and software that will be used to optimize its use, taking care of body language, and use a script to optimize the use of time, allowing the students to express themselves freely”

R=Thank you very much for your comments, the modifications have been made in the document.

The following sentence needs clarification: “knowledge transfer by teachers and found that most of the knowledge transfer by teachers” PAGE 3 Incorrect spelling: "stu-dents", “higher–education institutions”, “techno- logical”

R=Thank you very much for your comments, the modifications have been made in the document. See lines 51-58.

Please explain the differences between the types of higher education institutions, so that this information adds value to the text and to the reader;  Why did you choose TU and PU? What are the implications of this choice?  Please explain how the following objectives are connected to the research questions: Identify the socio–demographic aspects of the teachers that has an impact on the use of virtual learning platforms (VLPs); to know the perception of teaching staff on the higher education institutions (HEIs), both technological and polytechnical universities (TUs and PUs) over the use of VLP during the COVID-19 crisis;

Thank you very much, the modifications and clarifications have been made. The correct acronyms are Technological Universities (TUs) and Polytechnic Universities (PUs), which are institutions of higher education in Mexico. Adjustments were made throughout the document. The wording of the objectives was improved to connect with the problem definition and research questions. see lines 197-206.

Problem statement

The first research question needs to be much more specific (Question 1: How the pandemic affected teachers?) Related to “Question 2: Are there differences in the affectation by the pandemic between male and female?” - Are there no studies that answer this question?

How is your study different? Why does this question not appear in the research objectives?

Related to “This work intended to identify the voices and feelings of the inter-nal stakeholders of higher education,” – This is different of what you stated as your research objectives PAGE 6 Correct: “mexican case” Correct: “TUs an PUs”

R=Thank you very much, the modifications and clarifications have been made See lines 205-218.

The following sentences need clarification:

“performance with these said impacts (pressures)” “students instruction were modified because of the 30% theory, 70% practice distribution of Educational Model of these universities”

R=Thank you very much for the suggestion. See lines 234-237.

“Through the research, applied in 22 TUs and PUs during 2020, we were looking to analyze the impacts and obstacles teachers had in this paradigmatic change, as in the competencies developed by teachers, real challenges in the change from face-to-face to virtual classes, and the voice of the teachers.” – This is very confusing. Is it another research? Was it published? How this add to the text?

R=Thank you very much, the modifications have been made

“an important sector of higher education is integrated by the TUs and PUs” – How is this important? There is a need to explain to the reader, mainly for those that do not live in Mexico.

R=Thank you very much, the entire citation has been updated. See lines 99-105.

The following sentence is confusing: “In this sense and the possible unfavorable situations of the change of scheme in teaching and learning” Page 5

Consider the following paragraph: “The feeling of the professors of the TUs and PUs in Mexico, pointed out a series of impacts that were not favorable for their performance such as the economic impact and the investment made, their digital competences and the consequences in the formation of the students since the Educational Model of the TUs and PUs implies classes with 30% of theory and 70% of practice” – Where these data came from? There are repeated sentences here and in the Introduction

Missing space: “pointof”

R= See lines 99-105.

R=Thank you very much for the suggestion in the Problem statement section and the necessary changes have been made.

Methodology

In the explanation of step 2 there are two colons in a row

The figure must be self-explanatory; it is not possible to understand step 4, it is necessary to indicate in the figure that it is about data collection so as not to require the reader to make assumptions

The following sentence is confusing: “This study was carried out in teachers” Page 6

Consider the following: “The result obtained from the study was divided into five categories: academic, social, economic, emotional and health.” – Please explain why did you choose these categories and not others.

Consider the following: “The student measuring instrument of the TUs and PUs includes 25 items (questions) that describe the contextual environment of each of the categories” – Please explain how did you arrive in these 25 items? What were the methodological approaches adopted to ensure reliability that such questions describe what they should?

Consider the following: “The sample is not representative, but all subjects (teachers) in the population had the same probability of being part of the sample”

– This needs more explanation. If this sample is not representative, how representative is it? Consider the following: “Technological University of Tecamachalco has the highest participation with 26.9%, followed by the Technological University of Tamaulipas North with 10.5% and in third place is the Polytechnic University of Amozoc which has 7% representation.” – Please explain how this information add to the knowledge and how it contribute to answer the research questions and objectives

R= Thank you very much for your comment, the paragraph was deleted as it was not relevant.

This sentence is repeated several times throughout the text: “The study of the social, economic, academic, emotional and health effects caused by COVID-19 in students is focused on 15 TUs and 7 PUs” REMOVED FROM PAGE 8 GENERAL RESULTS, PARAGRAPH 2, LINE 5 These sentences need to be rewritten: “Question 2, gender. Male...”, “Question 3, teachers age range. 37% of the teachers”, “In Regards to question 13”, “Question 19, how do you feel when you are teaching virtual class? it is observed that”, many others – Please verify the entire results section for correcting spelling and punctuation In the previous sections you use the expression "teacher" and in the results you use "professor". This needs clarification as they have different meanings according to countries. The text needs to be clear so that the reader can understand what is being said.

R=Thank you very much for your suggestions, the necessary changes have been made in the methodology section. The wording of the research questions was adjusted See lines 215-218.

There are parts of the text that are not formatted accordingly There are words in red in the discussion

R=Thank you very much for your suggestions, the necessary changes have been made.

Reviewer 2 Report

The article is focusing on a subject that should be considered very important and very actual; technic universities have felt the most negative effects of the Covid-Pandemic changes imposed in higher education institutions. However, we are pointing bellow some observations and suggestions that could help the authors to improve the article. 

  • In the „Abstract” Section it would be very useful that the authors mention not only the acronym – MCA, but the entire syntagm: „multiple correspondence analysis”. This will help more the comprehension of the ideas presented in the „Abstract” section.
  • Also, in the „Abstract” section, the phrase : „Through a MCA analysis, it was determined that of the total number of teachers who responded to the questionnaire, 50% were comfortable with the online teaching model and 50% were not”, might be formulated in a different way, showing, of course, its main nuance – that the opinions on online education are strongly divided (the principle of the “double edge sword”). Our suggestion is that the authors should not mention the exact percentages (50% to 50%).
  • In the “Introduction” and “Problem statement” it would be more useful to use syntagms as “Fedock (2019)”, than “Fedock in 2019”.
  • Please specify more clearly the “Objectives” of the research (point 1.1.), because in the paragraph before the authors mention other objectives. Also, please use ‘Objectives”, instead of “Objective”, because the research has had more than one objective.
  • It might be very interesting if the “Research questions” could be transformed in “Research hypothesis” and if these hypotheses would be presented also in the 1.2. Paragraph (it could be renamed “Research questions and research hypothesis”)
  • Our impression is that the phrase “Through the research, applied in 22  TUs  and  PUs  during  2020,  we  were  looking  to  analyze  the  impacts  and  obstacles teachers had  in this paradigmatic change, as in the competencies developed by  teachers, real challenges in  the change from face -to-face to virtual classes, and the voice of the teachers” has a confuse meaning (maybe because of the bad English translation). Our suggestion is that this phrase should be formulated in a different view (page 4).
  • We appreciate the description of the 5 steps of the research conducted by the authors. However, if we are reading Step 5 we identify a confusion: if the research supposed a “survey tool”, this means that the article is presenting the results of a quantitative research, not a qualitative one. We cannot extract “qualitative aspects” (page 5), from quantitative research.
  • Also, the readers should be confused about the year in which the research was conducted: it was 2020 or 2021? Because the authors mention both years. It would be extremely useful to clarify this aspect.
  • We appreciate the large number of respondents, that were targetted by the empiric research, this being an atuu for the article, especially for the results of the research.
  • In the 4th section – „Results”., the first phrase – „The teachers from the TUs and PUs are an essential part of the educational system that carry out the teaching––learning process”, might be formulated in a different way
  • Section 4.2. – General results: please eliminate this phrase: Socio-demographic data including: participating Universities (Q1), teacher education program (Q2), gender (Q3) and age of teachers (Q3)”. It is simple description of the 3 (or 4???) questions. Is has no scientific relevance, the informations generate confusion, Q3 is mentioned twice, referring both to the age of teachers and to their sender”.
  • Overall, our suggestion for the “Results” part of the article is that the authors should avoid simple descriptions of each question; this might diminish the scientific importance of this part of the manuscript. This section requires MAJOR revision, in terms of written conclusions and English language proficiency (many paragraphs have no sense; we have faced difficulties in understanding the main idea)
  • Please avoid these simple phrases (our suggestion is to eliminate them), which are more appropriate for an oral presentation, not a scientific article. (example: “ Question 2, gender”; Question 3, teachers age range.)
  • Page 6 – “The surveys carried out were applied” – please use the singular form “the survey carried out” (it is about one method.)
  • Section 4.2. – General results: the phrase “The study of the social, economic, academic, emotional and health effects caused by COVID-19 in students is focused on 15 TUs and 7 PUs.” has no relevance here. Moreover, this aspect was already mentioned by the authors in the previous sections.
  • We appreciate the data presented in 4.3. Paragraph - “Statistical analysis”. For this paragraph, our suggestion it related to the English translation, because some parts seem to be mistranslated from the original language.
  • On page 15 – the phrase “The present work gives an initial view, especially on the first steps of the transition to teaching using digital devices in April 2021. Results reflect an educational community in great uncertainty regarding the process and the future of the period 2019/2020 and the confinement situation since March 2021” could generate confusion, in terms of events’ chronology (once again, the same question: it was a research conducted in 2020 or in 2021?)
  • We appreciate that in the last part of the article, the authors have indicated how the results obtained have answered to the research questions. If they will add also “research hypothesis”, it would be very useful to indicate here is the research hypotheses were validated or not validated.
  • We suggest changing the syntagm “health scope” “academic scope”, “social scope” with “health effects”, “social effects”, “academic effects” etc.
  • The topic is very interesting. The article could be published only after major revisions, to help the manuscript have a more scientific value. There are major errors in English language which can indicate the need of a more accurate and major revision of the manuscript.  

Author Response

Reviewer 2

In the „Abstract” Section it would be very useful that the authors mention not only the acronym – MCA, but the entire syntagm: „multiple correspondence analysis”. This will help more the comprehension of the ideas presented in the „Abstract” section.

R= Thank you very much for your comments, the correction was made in the abstract. See abstract line 13. Added section 4.3 see lines 324-335.

Also, in the „Abstract” section, the phrase : „Through a MCA analysis, it was determined that of the total number of teachers who responded to the questionnaire, 50% were comfortable with the online teaching model and 50% were not”, might be formulated in a different way, showing, of course, its main nuance – that the opinions on online education are strongly divided (the principle of the “double edge sword”). Our suggestion is that the authors should not mention the exact percentages (50% to 50%).

R=Thank you very much for your suggestions, the necessary changes have been made

In the “Introduction” and “Problem statement” it would be more useful to use syntagms as

“Fedock (2019)”, than “Fedock in 2019”.

R=Thank you very much for your suggestions, the necessary changes have been made.

Please specify more clearly the “Objectives” of the research (point 1.1.), because in the paragraph before the authors mention other objectives. Also, please use ‘Objectives”, instead of “Objective”, because the research has had more than one objective. 

It might be very interesting if the “Research questions” could be transformed in “Research hypothesis” and if these hypotheses would be presented also in the 1.2. Paragraph (it could be renamed

R=Thank you very much for your suggestions, the necessary changes have been made. See lines 197-204.

Our impression is that the phrase “Through the research, applied in 22 TUs and PUs during 2020, we were looking to analyze the impacts and obstacles teachers had in this paradigmatic change, as in the competencies developed by teachers, real challenges in the change from face -to-face to virtual classes, and the voice of the teachers” has a confuse meaning (maybe because of the bad English translation). Our suggestion is that this phrase should be formulated in a different view (page 4).

R=Thank you very much for your suggestions, the necessary changes have been made.

We appreciate the description of the 5 steps of the research conducted by the authors. However, if we are reading Step 5 we identify a confusion: if the research supposed a “survey tool”, this means that the article is presenting the results of a quantitative research, not a qualitative one. We cannot extract “qualitative aspects” (page 5), from quantitative research.

R=Thank you very much for your suggestions, the necessary changes have been made. See line 285.

Also, the readers should be confused about the year in which the research was conducted: it was 2020 or 2021? Because the authors mention both years. It would be extremely useful to clarify this aspect.

R=Thank you very much for your suggestions, the necessary changes have been made.

We appreciate the large number of respondents, that were targetted by the empiric research, this being an atuu for the article, especially for the results of the research.

In the 4th section – „Results”., the first phrase – „The teachers from the TUs and PUs are an essential part of the educational system that carry out the teaching––learning process”, might be formulated in a different way

R=Thank you very much for your suggestions, the necessary changes have been made. See line 338

Section 4.2. – General results: please eliminate this phrase: Socio- demographic data including: participating Universities (Q1), teacher education program (Q2), gender (Q3) and age of teachers (Q3)”. It is simple description of the 3 (or 4???) questions. Is has no scientific relevance, the informations generate confusion, Q3 is mentioned twice, referring both to the age of teachers and to their sender”.

R=Thank you very much for your suggestions, the necessary changes have been made. See lines 342-354.

Overall, our suggestion for the “Results” part of the article is that the authors should avoid simple descriptions of each question; this might diminish the scientific importance of this part of the manuscript. This section requires MAJOR revision, in terms of written conclusions and English language proficiency (many paragraphs have no sense; we have faced difficulties in understanding the main idea)

R=Thank you very much for your comments, a native speaker did the language review.

Please avoid these simple phrases (our suggestion is to eliminate them), which are more appropriate for an oral presentation, not a scientific article. (example: “ Question 2, gender”; Question 3, teachers age range.)

Page 6 – “The surveys carried out were applied” – please use the singular form “the survey carried out” (it is about one method.)  Section 4.2. – General results: the phrase “The study of the social, economic, academic, emotional and health effects caused by COVID-19 in students is focused on 15 TUs and 7 PUs.” has no relevance here. Moreover, this aspect was already mentioned by the authors in the previous sections.

R=Thank you very much for your suggestions, the necessary changes have been made, See lines 342-354.

We appreciate the data presented in 4.3. Paragraph - “Statistical analysis”. For this paragraph, our suggestion it related to the English translation, because some parts seem to be mistranslated from the original language.

R=Thank you very much for your suggestions, the necessary changes have been made

On page 15 – the phrase “The present work gives an initial view, especially on the first steps of the transition to teaching using digital devices in April 2021. Results reflect an educational community in great uncertainty regarding the process and the future of the period 2019/2020 and the confinement situation since March 2021” could generate confusion, in terms of events’ chronology (once again, the same question: it was a research conducted in 2020 or in 2021?)

R=Thank you very much for your comments, the time period has been corrected, the research was conducted in 2021.

We appreciate that in the last part of the article, the authors have indicated how the results obtained have answered to the research questions. If they will add also “research hypothesis”, it would be very useful to indicate here is the research hypotheses were validated or not validated. We suggest changing the syntagm “health scope” “academic scope”, “social scope” with “health effects”, “social effects”, “academic effects” etc.

R=Thank you very much for your suggestion, in the conclusions section the suggestion has been taken care of.

The topic is very interesting. The article could be published only

after major revisions, to help the manuscript have a more scientific value. There are major errors in English language which can indicate the need of a more accurate and major revision of the manuscript.

R=Thank you very much for the comment A native speaker made the language correction.

Reviewer 3 Report

The authors have tried to bring a serious work with this paper. There are a few imperfections those must be improved:

In the introduction, originality must be justified in the context of previous studies. Besides, it is advisable to state clearly the aim of this paper and justify the theoretical framework. The importance of research does not arise. Results presentation section seems to be general. Further, what is the major contribution of this study? Try to illustrate them in the manuscript. The results should be correlated with the research questions in the discussion section. The conclusions drawn are not well justified in the data collected. Also, conclusions should be rewritten to understand the importance of research. Are there any further research streams? All figures should be revised to be clear. Authors should follow the format of the journal.

Author Response

Reviewer 3

In the introduction, originality must be justified in the context of previous studies. Besides, it is advisable to state clearly the aim of this paper and justify the theoretical framework. The importance of research does not arise.

R=Thank you very much for your suggestions, the necessary changes have been made. See lines 140-196.

Results presentation section seems to be general. Further, what is the major contribution of this study? Try to illustrate them in the manuscript.

R=Thank you very much for your suggestions, the necessary changes have been made. See lines 534-601. Lines 776-867.

 The results should be correlated with the research questions in the discussion section.

R=Thank you very much for your suggestions, the necessary changes have been made.  See lines 792-823

The conclusions drawn are not well justified in the data collected. Also, conclusions should be rewritten to understand the importance of research. Are there any further research streams? All figures should be revised to be clear. Authors should follow the format of the journal.

R=Thank you very much for your suggestions, the necessary changes have been made

Round 2

Reviewer 1 Report

Dear author(s),

Thanks for the new version of your article.

I'm afraid there are serious flaws that negatively affect the contribution, reliability and validity of the paper. I am afraid that there are serious flaws that negatively affect the contribution, reliability and validity of the paper. Here are some examples, but they are present throughout the text.

1) Confusing sentences and paragraphs

For instance, right in the abstract it is stated that "By means of a G-test, it was determined that there are no factors that have a positive or negative effect on the gender of the teachers." – How can any of the factors studied affect a person's gender?

2) Weak theoretical basis

For instance, gender is a central issue of the study, but there is no solid discussion on the topic.

3) Questionable methodology and results, without theoretical and scientific arguments

In the conclusion it is stated that "Hypothesis H1: There are differences in the impact of the pandemic between male and female, was discarded.", referring. This is completely contrary to what the most important scientific references say, as well as international bodies such as the United Nations, and there is no discussion or theoretical basis on this.

4) Language

The structure of the text and the writing itself (English language) make reading and understanding very difficult.

Thank you for the opportunity to try to contribute to your work.

Author Response

Thank you very much for your contributions

1) Confusing sentences and paragraphs

For instance, right in the abstract it is stated that "By means of a G-test, it was determined that there are no factors that have a positive or negative effect on the gender of the teachers." – How can any of the factors studied affect a person's gender?

R= Thank you very much for your comment. We have made the corrections. See Lines 11-16

2) Weak theoretical basis

For instance, gender is a central issue of the study, but there is no solid discussion on the topic.

R= Thank you very much for your comments. 

We have included a specific section on gender. See lines 107-246

We have addressed the issue in the discussion. See lines 601-855

3) Questionable methodology and results, without theoretical and scientific arguments.

In the conclusion it is stated that "Hypothesis H1: There are differences in the impact of the pandemic between male and female, was discarded.", referring. This is completely contrary to what the most important scientific references say, as well as international bodies such as the United Nations, and there is no discussion or theoretical basis on this.

R=Thank you very much for your comments

We have made the necessary corrections. See lines  592-604, 895-903

4) Language

The structure of the text and the writing itself (English language) make reading and understanding very difficult.

Thank you very much for your comments.

The language revision service offered by the Publisher id 41498 has been contracted.

Reviewer 2 Report

The article was improved according to the suggestions from the first review report. The authors have added research hypotheses, which will increase the scientific value of the manuscript. In the section in which they are presenting the methods, they have introduced a new paragraph describing the multiple corresponding analysis, which is also improving the manuscript. 

In the actual form, after the modifications made, the manuscript could be published 

Author Response

Reviewer  2

The article was improved according to the suggestions from the first review report. The authors have added research hypotheses, which will increase the scientific value of the manuscript. In the section in which they are presenting the methods, they have introduced a new paragraph describing the multiple corresponding analysis, which is also improving the manuscript.  In the actual form, after the modifications made, the manuscript could be published 

R=Thank you very much for your review and input.  All the suggested improvements have been made.

Reviewer 3 Report

The authors responded to my comments.

Author Response

Reviewer  3

The authors responded to my comments.

R= Thank you very much for your review and input.  All the suggested improvements have been made.

Round 3

Reviewer 1 Report

Dear author(s),

Thank you for the improved version of your manuscript.

Please consider the following:

  • All the references are as [?] – it is not possible to evaluate
  • All figures are “??”
  • The same for “Table Appendix ??”
  • In the heading is informed the following: “Version March 28, 2022 submitted to Micromachines 13 of ??” - This is not the right journal
  • There is a need to review the text. For example in lines 597-600:

"H1: Are there differences in the affectation by the pandemic between males and females. The Hypothesis is not rejected but due to results and statistical analysis it was determined that the main affectations902 were economical, the equipment and connectivity."

  • H1 is a statement or a question? What is “affectations902”?
  • Another example is in line 605: “5.3.2. MCA analisys” – There is a need to fix “analysis”, but the entire text needs to be reviewed.

Author Response

Reviewer 1

  • The same for “Table Appendix ??”
  • All the references are as [?] – it is not possible to evaluate. All figures are “??”

Thank you very much, corrections were made in all details.

  • In the heading is informed the following: “Version March 28, 2022 submitted to Micromachines 13 of ??” - This is not the right journal

Thank you very much, corrections were made in all details.

  • There is a need to review the text. For example in lines 597-600:
  • "H1: Are there differences in the affectation by the pandemic between males and females. The Hypothesis is not rejected but due to results and statistical analysis it was determined that the main affectations902 were economical, the equipment and connectivity."

Thank you very much, corrections were made in all details. See lines  597

  • H1 is a statement or a question? What is “affectations902”?
  • Another example is in line 605: “5.3.2. MCA analisys” – There is a need to fix “analysis”, but the entire text needs to be reviewed.

Thank you very much, corrections were made in all details. See lines 597-605

 The English Editing service was given by mdpi according to your suggestions.
